

# Modelling of sediment transport and morphological evolution under the combined action of waves and currents

Guilherme Franz[1], Matthias T. Delpey[2], David Brito[3], Lígia Pinto[1], Paulo Leitão[4], Ramiro Neves[1]

[1]MARETEC, Instituto Superior Técnico, Universidade de Lisboa, Av. Rovisco Pais, 1049-001, Lisboa, Portugal
[2]Centre Rivages Pro Tech, SUEZ, 2 allée Théodore Monod, Bidart, France
[3]ACTION MODULERS, Estrada Principal, n°29, Paz, 2640-583 Mafra, Portugal
[4]HIDROMOD, Rua Rui Teles Palhinha, n°4, 1°, Leião, 2740-278 Porto Salvo, Portugal

*Correspondence to*: Guilherme Franz (guilherme.franz@tecnico.ulisboa.pt)

**Abstract.** Coastal defence structures are often constructed to prevent beach erosion. However, poorly designed structures may cause serious erosion problems in the downdrift direction. Morphological models are useful tools to predict such impacts and assess the efficiency of defence structures for different scenarios. Nevertheless, morphological modelling is still a topic under intense research effort. The processes simulated by a morphological model depend on model complexity. For instance, undertow currents are neglected in coastal area models (2DH), which is a limitation for simulating the evolution of beach profiles for long periods. Model limitations are generally overcome by predefining invariant equilibrium profiles that are allowed to shift offshore or onshore. A more flexible approach is described in this paper, which can be generalised to 3D models. The present work is based on the coupling of the MOHID modelling system and SWAN wave model. The impacts of different designs of detached breakwaters and groynes were simulated in a schematic beach configuration following a 2DH approach. The results of bathymetry evolution are in agreement with the patterns found in the literature for several existing structures. The model was also tested in a 3D test case to simulate the formation of sandbars by undertow currents. The findings of this work confirmed the applicability of the MOHID modelling system to study sediment transport and morphological changes in coastal zones under the combined action of waves and currents. The same modelling methodology was applied to a coastal zone (Costa da Caparica) located at the mouth of a mesotidal estuary (Tagus Estuary, Portugal) to evaluate the hydrodynamics and sediment transport in calm water conditions and during events of highly energetic waves.

## 1 Introduction

The morphological features of the coastal zone depend on the sediment characteristics and the combined action of waves and currents. Wind waves are the main energy source for most beaches. Particularly in the surf zone, waves may induce considerable changes in mean sea level and strong currents (Longuet-Higgins, 1970a, 1970b, 1983). The pattern of surf zone currents varies with the obliquity of waves approaching the shore as well as with bathymetric heterogeneities, leading to longshore and rip currents. Sediment is usually carried shoreward during low wave conditions, mainly due to the asymmetry



of waves in shallow waters (Myrhaug et al., 2004). The sediment accumulated during these periods may be eroded very rapidly under high wave conditions during a major storm. Following these energetic events, the bottom profile may recover its initial shape only if the longshore transport of sediment during the storm was low. Different structures such as breakwaters and groynes can prevent the along-coast movement of sediment (e.g., Dally and Pope, 1986). However, serious

erosion problems in the downdrift direction may arise from the construction of these structures. Morphological models are useful tools to assess the impact of defence structures, enabling us to consider different wave conditions and structure designs.

The complexity of morphological models ranges from coastal profile models to bi- or three-dimensional models. Actually, morphological models are a set of different models or modules, depending on the chosen approach. Here the focus is set on

the coupling of a spectral wave model with a phase-averaged hydrodynamic and sediment transport model. Spectral wave models offer a representation of the physical processes related to the generation, propagation and dissipation of waves (e.g., Booij et al., 1999). The wave-induced forces computed by a wave model can be provided for a hydrodynamic model in order to simulate wave-related phenomena, such as wave set-up, wave-induced currents and mixing. On the other hand, the hydrodynamic model can return water levels and currents for the wave model (e.g., Warner et al., 2008). Additional

processes can be considered in the hydrodynamic model, such as wind action, tidal motion, and river discharges. The transport of suspended sediment may be simulated by an advection/diffusion model. The mechanism of erosion/deposition of sediments is controlled by the bed shear stress induced by currents and waves. The bathymetry evolution resulting from the total sediment transport (suspended load and bed load) affects the patterns of currents and waves. Due to the interdependence of the physical processes involved in sediment dynamics, all of these models must be coupled.

The hydrodynamics inside the surf zone is influenced by important three-dimensional effects. Due to the absence of three-dimensional processes (e.g., undertow), coastal area models (2DH) fail to reproduce a consistent evolution of beach profiles for long periods. This shortcoming can be overcome by predefining invariant equilibrium profiles. In this case, the equilibrium profile only shifts offshore or onshore depending on the overall sediment balance along the profile, similarly to a coastal profile model (e.g., Kriebel and Dean, 1985; Kristensen et al., 2013). As the sediment transport in the swash zone is

usually neglected in large-scale 2DH models, this approach also has the advantage of updating the shoreline position. However, limitations arise when a structure is present in the surf zone. Moreover, the processes responsible for sandbar evolution are not considered. Attempts to simulate sandbar dynamics have been performed generally with cross-shore bi-dimensional (2DV) and quasi-3D models (e.g., Drønen and Deigaard, 2007; Ruessink et al., 2007). Nevertheless, the proper reproduction of sandbar migration is still an active topic of research (e.g., Dubarbier et al., 2015). Inaccuracies in the cross-

shore sediment transport may degrade the coastal profile, which is a restriction for the simulation of long-term morphological evolution.

In this work, we test a more flexible approach to overcome the 2DH model limitations in order to simulate morphological evolution for long periods. Instead of fixing an equilibrium profile to update the bathymetry and shoreline position, we defined a maximum slope that when surpassed generates sediment transport in the downslope direction. Thus, rather than



extrapolating erosion or deposition fluxes over the entire profile, only individual grid cells are affected. This approach may be more appropriate in order to consider the effect of non-uniform grain-size distributions on the overall sediment transport, through morphological models that account for multiple sediment fractions. Grain-size sorting is generally observed along the cross-shore beach profile, as well as in the longshore beach direction (Komar, 1998). Furthermore, the method can be

generalised to 3D models for a better representation of sandbar slopes and to update the shoreline position.

This paper is divided into five sections. A brief description of the effect of waves in the nearshore hydrodynamics and sediment transport is given in Sect. 2. The numerical modelling approach is presented in Sect. 3. The methodology was verified for different test cases (Sect. 4). Firstly, the morphological evolution of a schematic beach was evaluated for different designs of coastal defence structures (detached breakwaters and groynes) following a 2DH model configuration.

The model was applied later in a 3D configuration for the same schematic beach to verify the development of sandbars. Finally, the numerical modelling methodology was applied to assess the hydrodynamics and sediment transport under extreme wave conditions in a coastal zone (Costa da Caparica) located at the mouth of a mesotidal estuary (Tagus Estuary, Portugal). The main conclusions found from these test cases are discussed in Sect. 5.

## 2 Background

The effect of breaking waves on the mean sea level (wave set-up) is known since the laboratory measurements performed by Saville (1961), confirmed further by Bowen et al. (1968). This tilt of the mean sea level is explained by the horizontal flux of momentum carried by waves, or equivalently by the radiation stress, a vertically integrated momentum flux whose gradient balances the wave set-up (Longuet-Higgins and Stewart 1962, 1964). In the surf zone, wave heights and orbital velocities decrease towards the shore due to wave breaking. As a consequence, the radiation stress also decreases, resulting in a force

directed towards the shore. This force is balanced by a hydrostatic pressure gradient that increases the mean sea level onshore.

Breaking waves can also drive strong currents in the surf zone, which are important for sediment transport and morphological evolution in the coastal zone. The horizontal mass transport associated with waves, or Stokes drift, is oriented shoreward and vertically sheared, being more intense at the surface (e.g. Ardhuin et al., 2008). As a result, mass conservation

in the nearshore is satisfied by a seaward transport in the lower part of the water column, called undertow, which has an important role in sandbar formation. The undertow is strongest in steep beaches and may be insignificant for moderate beach slopes, where circulation tends to break up into rip currents (Longuet-Higgins, 1983). Also, obliquely breaking waves generate longshore currents (Longuet-Higgins, 1970a, 1970b) and, consequently, longshore sediment transport. Although nearshore sediment dynamics are dominated by wave action, tidal motion can also play an important role, alternately moving

the breaker zone and shoreline position shoreward and seaward, which may prevent the development of longshore bars in the surf zone (e.g., Levoy et al., 2000).



## 3 Numerical model

The present work is based on the coupling of the MOHID modelling system (Leitão, 2003; Leitão et al., 2008) and the SWAN wave model (Booij et al., 1999). The MOHID code organisation follows an object oriented strategy that permits the integration of different scales and processes. Herein, the focus is given to the hydrodynamics and sediment transport in the

nearshore area. Thus, we considered the processes related to the hydrodynamics, turbulence, advection/diffusion of suspended sediment, erosion/deposition of sediments, bed load sediment transport and morphological evolution. A brief description of the most important aspects of the hydrodynamic model for this work is presented in this section, followed by the main novelties implemented in the MOHID code: a new method to calculate the bed load transport under the combined effect of currents and waves; and a bed slope correction considered to overcome the 2DH model limitations and to update the

shoreline.

The SWAN wave model represents the processes of wave generation, propagation, refraction, shoaling, nonlinear (quadruplet and triad) wave-wave interactions, and dissipation (whitecapping, bottom friction, and depth-induced breaking). More information can be found in the documentation of the SWAN wave model (http://swanmodel.sourceforge.net/). Both MOHID and SWAN can be run with multiple processors using shared (OpenMP) or distributed (OpenMPI) memory

architectures. In this work, we implemented MPI directives in the MOHID module responsible for the calculations of bed load transport and morphological evolution (Sediment module), which can consider multiple sediment classes. In addition to speeding up morphological changes by an acceleration factor, this new development reduces the computational time required for modelling bed evolution over long time periods.

### 3.1 Hydrodynamic module

The MOHID hydrodynamic module solves the Navier-Stokes equations, considering the hydrostatic, Boussinesq and Reynolds approximations (Martins, 2000, Leitão, 2003):

$$\frac{\partial}{\partial t}\int_V \vec{v}_H \, dV = -\oint_A \vec{v}_H(\vec{v}.\vec{n}) \, dA + \oint_A \upsilon_T\big(\vec{\nabla}(\vec{v}_H).\vec{n}\big) \, dA - \frac{1}{\rho}\oint_A p.\vec{n}_H \, dA + \int_V 2\vec{\Omega}\times\vec{v}_H dV + \vec{F} \tag{1}$$

where $V$ represents the control volume, $\vec{v}_H = (u, v)$ the horizontal velocity vector, $\vec{v} = (u, v, w)$ the velocity vector, $\vec{n}$ the normal vector to the bounding surface ($A$), $\vec{n}_H$ the normal vector related to the horizontal plane, $\upsilon_T$ the turbulent viscosity, $\rho$

the water density, $p = g\int_z^\eta \rho dz + p_{atm}$ the water pressure, $g$ the gravitational acceleration, $p_{atm}$ the atmospheric pressure, $\eta$ the water level, $\vec{\Omega}$ the earth rotation vector, and $\vec{F}$ the external forces, which include the wave-induced force (gradient of the radiation stress) computed by the wave model. The wave-induced force was considered in the MOHID hydrodynamic module in previous studies to compute the effects of waves on sea level (Malhadas et al., 2009), water renewal in a coastal lagoon (Malhadas et al., 2010), and coastal water dispersion in an estuarine bay (Delpey et al., 2014).



The spatial discretization is performed by following the finite volume method. The water level and vertical velocity are computed through the continuity equation integrated over the entire water column or applied to each control volume, respectively. The equations are solved through the Alternating Direction Implicit (ADI) method in an Arakawa C-grid structure. A generic vertical discretization allows implementing different types of vertical coordinates (e.g, Sigma or Cartesian) (Martins, 2001). The turbulent viscosity is computed differently for the horizontal and vertical directions. The horizontal turbulent viscosity is defined as a constant value, based on the grid resolution and a reference velocity, or as a function of horizontal velocity gradients, based on Smagorinsky (1963). The vertical turbulent viscosity is computed by the Global Ocean Turbulence Model (GOTM), which is coupled to MOHID and consists of a set of turbulence-closure models (Buchard et al., 1999; Villarreal et al., 2005).

To solve the Navier-Stokes and continuity equation, appropriate boundary conditions are required for the lateral (e.g., land and open sea), surface and bottom boundaries. MOHID has the option of a great variety of open boundary conditions of several types: Dirichlet, Neumann, radiation, cyclic, relaxation (or nudging), etc. Some boundary conditions can be combinations of the types enumerated, e.g., a combination of radiation with nudging (Blumberg and Kantha, 1985). Open boundary conditions (OBC) can be imposed by prescribing the values of a specific variable (Dirichlet boundary condition). This condition is commonly applied in coastal models to impose tidal levels when the correspondent barotropic velocities are not available. On the other hand, following a Neumann boundary condition, the gradient of a specific variable is imposed instead of a prescribed value. Assuming a null gradient condition, the value of a variable at a boundary point is equal to the value at an adjacent interior point. When the shoreline location and bathymetry are uniforms, e.g, in schematic cases, cyclic boundary conditions can be applied.

A relaxation scheme can be applied as an OBC by assuming a decay time that increases gradually from the boundary to infinite after a defined number of cells (see Martinsen and Engedahl, 1987; Engedahl, 1995):

$$P^{t+\Delta t} = P^* + (P^{ext} - P^*)\frac{\Delta t}{T_d} \tag{2}$$

where $P$ is a generic property, $P^*$ is the property value calculated by the model, $P^{ext}$ is the reference value of the property, $\Delta t$ is the model time step and $T_d$ is the relaxation time scale.

Radiation methods can also be used to impose the OBC, which allow the propagation of internal disturbances on water levels through the open boundaries. These disturbances can be caused, for example, by the wave forces. MOHID has two types of radiation conditions (Leitão, 2003), based on Blumberg and Kantha (1985), Eq. (3), and Flather (1976), Eq. (4):

$$\frac{\partial \eta}{\partial t} + (\overrightarrow{C_E}.\vec{n})\Delta \eta = \frac{1}{T_d}(\eta_{ext} - \eta) \tag{3}$$

$$q - q_{ext} = (\eta_{ext} - \eta)(\overrightarrow{C_E}.\vec{n}) \tag{4}$$



where $\eta$ and $q$ are the water level and specific flow, respectively; $\eta_{ext}$ and $q_{ext}$ are the imposed values of $\eta$ and $q$ at open boundary points; $\overrightarrow{C_E}$ is the celerity of internal water level disturbances or the celerity of external waves ($C_E = \sqrt{gh}$). When $T_d$ is approximated to infinity in the Eq. (3), the OBC becomes totally passive, which means that the water levels at the boundary points are computed only from the internal water levels. An approximated null relaxation time scale means that the water level is imposed as $\eta_{ext}$. The Flather radiation condition is mostly used in nested model domains, when velocities and water levels are known at the open boundaries. Many of the OBC used by MOHID considered the concept of an external (or reference) solution. These solutions can be provided via input file or via a one-way nesting of a chain of models. This last process is used to downscale from large-scale domains to local ones (e.g., Franz et al., 2016). In the case of land points, the closed boundary condition is imposed as null fluxes of mass and momentum in the perpendicular direction. The covering and uncovering of boundary cells can be represented in MOHID by a wetting/drying scheme (Martins et al. (2001).

At the surface, fluxes of momentum from wind action and wave breaking can be considered (Delpey et al., 2014). At the bottom, the method proposed by Soulsby and Clarke (2005) to compute the bed shear stress was implemented in this work, consisting of a steady component due to currents together with an oscillatory component due to waves. In a laminar flow, the combined bed shear stress is a simple linear addition of the laminar current-alone and wave-alone shear stresses. However, in turbulent flows this addition is non-linear and the mean and oscillatory components of the stress are enhanced beyond the values of the laminar case. The mean bed shear stress is used for determining the friction governing the current, whereas the maximum shear stress is used to determine the threshold of sediment motion. The turbulence generated by the skin friction acts directly on bottom sediment grains (Einstein, 1950), contrarily to that related to bed forms. Thus, the threshold of sediment motion depends on the grain-related bed shear stress.

## 3.2 Bed load transport

The bed load transport under the combined effect of currents and waves is computed following the semi-empirical formulation of Soulsby and Damgaard (2005). The formulation was derived for current plus sinusoidal and asymmetrical waves, as well as asymmetrical waves alone. Amoudry and Liu (2010) obtained a generally good agreement comparing the results of Soulsby and Damgaard (2005) formulations with a sheet flow model, concluding that it can be implemented in both intrawave and wave-averaged models in order to study sediment transport. The parallel $\Phi_\parallel$ and normal $\Phi_\perp$ components of the non-dimensional bed load transport vector in relation to the current direction are:

$$\Phi_\parallel = max\left(\Phi_{\parallel 1}, \Phi_{\parallel 2}\right) ; \text{ if } \theta_{max} > \theta_{cr} \tag{5a}$$

$$\Phi_{\parallel 1} = k_{\Phi 1}\theta_m^{k_{\Phi 2}}\left(\theta_m - \theta_{cr}\right)^{k_{\Phi 3}} \tag{5b}$$

$$\Phi_{\parallel 2} = k_{\Phi 1}(0.9534 + 0.1904\cos 2\emptyset)\theta_w^{1/2}\theta_m + k_{\Phi 1}\left(0.229\nabla\theta_w^{3/2}\cos\emptyset\right) \tag{5c}$$





$$\Phi_\perp = k_{\Phi 1} \frac{(0.1907\theta_w^2)}{\theta_w^{3/2} + (3/2)\theta_m^{3/2}} \left(\theta_m \sin 2\emptyset + 1.2\nabla\,\theta_w\,\sin\emptyset\right); \text{ if } \theta_{max} > \theta_{cr} \tag{6}$$

where $\theta_m$ and $\theta_w$ are the time-mean and oscillatory part of the bed shear stress (non-dimensional), respectively; $k_{\Phi 1}$, $k_{\Phi 2}$ and $k_{\Phi 3}$ are calibration coefficients that allow to represent different equations for the bed load transport found in the literature (e.g., Amoudry and Souza, 2011); $\emptyset$ is the angle between the wave propagation and current direction; $\nabla$ is a factor

that represents the wave's asymmetry; $\theta_{cr}$ is the critical non-dimensional bed shear stress, which depends on sediment diameter and bed-material gradation. The bed load transport is null ($\Phi_\parallel = \Phi_\perp = 0$) if $\theta_{cr}$ is greater than or equal to the maximum non-dimensional bed shear stress ($\theta_{max}$):

$$\theta_{max} = max\left(\theta_{max1}, \theta_{max2}\right) \tag{7a}$$

$$\theta_{max1} = \left(\left(\theta_m + \theta_w\,(1 + \nabla)\cos\emptyset\right)^2 + \left(\theta_w\,(1 + \nabla)\sin\emptyset\right)^2\right)^{1/2} \tag{7b}$$

$$\theta_{max2} = \left(\left(\theta_m + \theta_w\,(1 - \nabla)\cos(\emptyset + \pi)\right)^2 + \left(\theta_w\,(1 - \nabla)\sin(\emptyset + \pi)\right)^2\right)^{1/2} \tag{7c}$$

The bed load transport vector, $\vec{\Phi} = (\Phi_\parallel, \Phi_\perp)$, is enhanced in the presence of waves. Regarding the symmetrical case

($\nabla = 0$), the effect of the wave's asymmetry results in an additional increase in the normal component ($\Phi_\perp$), whereas the parallel component ($\Phi_\parallel$) can be increased or reduced, depending on the angle ($\emptyset$) between the wave propagation and current direction. As the water depth decreases, the wave's asymmetry becomes more significant. The asymmetry factor ($\nabla = \theta_{w,2}/\theta_{w,1}$) is defined as the ratio between the bed shear stress due to the wave's second harmonic ($\theta_{w,2}$) and basic harmonic ($\theta_{w,1}$), set to a maximum value of 0.2 (Soulsby and Damgaard, 2005). Considering the quadratic friction law ($\vec{\tau}_b = $

$\rho C_D \vec{v}_H |\vec{v}_H|$) to determine the magnitude of the wave's bed shear stress, the asymmetry factor is computed as:

$$\nabla = \left(\frac{U_{w,2}}{U_{w,1}}\right)^2 = \left(\frac{3}{4}\frac{\pi H_w}{L_w \sinh^3(k_w h)}\right)^2 \tag{8}$$

where $U_{w,2}$ and $U_{w,1}$ are the amplitude of near-bed wave-orbital velocity for the second harmonic and basic harmonic of Stokes second order wave theory (e.g., Greenwood and Davis, 2011), respectively; $H_w$ is the wave height; $L_w$ is the wavelength; $k_w = 2\pi/L_w$ is the wave number; and $h$ is the water depth.

To compute the sediment fluxes between grid cells, the components of the non-dimensional bed load transport vector are rotated to the grid referential (u-axis and v-axis):

$$\Phi_u = \Phi_\parallel \cos\emptyset_c - \Phi_\perp \sin\emptyset_c \tag{9a}$$



$$\Phi_v = \Phi_{\parallel} \sin \emptyset_c + \Phi_{\perp} \cos \emptyset_c \tag{9b}$$

where $\emptyset_c$ is the angle between the horizontal velocity vector and the u-axis. Thus, the bed load transport vector, $\vec{q} = (q_u, q_v)$, in mass units (kg m$^{-1}$s$^{-1}$) is equal to:

$$q_u = \rho_s \Phi_u \left[ g(\rho_r - 1)d^3 \right]^{1/2} \tag{10a}$$

$$q_v = \rho_s \Phi_v \left[ g(\rho_r - 1)d^3 \right]^{1/2} \tag{10b}$$

5    where $\rho_s$ is the sand particle density (kg m$^{-3}$); $\rho_r$ is the relative density ($\rho_s/\rho$); and $d$ is the sand representative diameter (m).

### 3.3 Bed slope correction

Wave action induces a shoreward sediment transport that has no counterpart in 2DH models, leading to sand accumulation in the nearshore and increasing the steepness of the beach profile. Actually, undertow currents are responsible for a seaward sediment transport, which may generate sandbars. Diverse opposing forces are responsible for creating an equilibrium

10   profile, which depends on sediment characteristics and wave heights (Dean, 1991). To account for the neglected forces in 2DH models, we defined a maximum slope ($\alpha_{max}$) that when exceeded induces sediment transport in the down slope direction. This artificial sediment transport may act as the undertow, transporting sediment seaward. The mass of sand ($M$) in the sediment column and, consequently, the bathymetry are updated when the bottom slope ($\alpha$) is larger than $\alpha_{max}$:

$$M_{i,j}^{t+1} = M_{i,j}^t - \Delta M \tag{11a}$$

$$M_{i,j+1}^{t+1} = M_{i,j+1}^t + \Delta M \tag{11b}$$

15   where:

$$\Delta M = \Delta z_b A \rho_s (1 - n), \alpha > \alpha_{max} \tag{12a}$$

$$\Delta M = 0, \alpha \leq \alpha_{max} \tag{12b}$$

in which $n$ is the sediment porosity, $A$ is the grid cell area, $t$ is an index symbol for time, and $i, j$ are index symbols to identify the grid cell ($i$ - line number, $j$ - column number). Considering the u-direction, the bed change in one time step ($\Delta z_b$) is computed as:

$$\Delta z_b = min\left( (|\alpha| - \alpha_{max})\Delta x, \Delta z_{b\,max} \right), \alpha > 0 \tag{13a}$$

$$\Delta z_b = min\left( -(|\alpha| - \alpha_{max})\Delta x, -\Delta z_{b\,max} \right), \alpha < 0 \tag{13b}$$



where $\alpha = \left(z_{b_{i,j+1}} - z_{b_{i,j}}\right)/\Delta x$, $\Delta x$ is the cell width, $z_b$ is the distance from the bed to a reference height (e.g., the hydrographic zero), and $\Delta z_{b_{max}}$ is a threshold to avoid numerical instabilities due to large shockwaves. Similar equations are used in the v-direction. Different values of the maximum slope ($\alpha_{max}$) can be defined in wet and dry cells. This method is based on that of Roelvink et al. (2009), previously applied to simulate dune erosion. The shoreline position is also updated
following this approach.

### 3.4 Model Coupling

The coupling between the MOHID modelling system and the SWAN wave model was performed through tools developed in the Fortran language as part of the MOHID code in order to convert the results to the appropriate format. An external tool was also developed in Python language to automatically manage the runs of the tools and models. Fields of significant wave
height, wave period, wavelength, wave direction, wave-induced force (radiation stress), and maximal orbital velocity near the bottom can be provided by SWAN to MOHID. In return, fields of water level, current velocity and bathymetry can be provided by MOHID to SWAN. The frequency of fields updating can be defined by the user for each application, depending on the variability of forcing conditions and speed of morphological changes.

## 4 Test Cases

### 4.1 Coastal Defence Structures

The morphological evolution of a schematic beach was simulated considering different designs of detached breakwaters and groynes to assess model results. Constant wave conditions were defined along the offshore boundary (1.5 m of wave height, 8 s of peak wave period and 15° of peak wave direction), following the JONSWAP spectrum. The hydrodynamic model was applied in 2DH mode, considering the vertically integrated wave-induced forces. The sand granulometry was uniform with a
diameter of 0.2 mm.

The MOHID domain was defined as 2 km cross-shore by 3 km alongshore, whereas the SWAN domain was defined as 2 km cross-shore by 9 km alongshore (3 km larger in each side of the MOHID domain). The grid resolution was equal in both models, ranging from 50 m x 50 m to 10 m x 10 m. A larger domain for the wave model was considered to avoid inaccuracies in the lateral boundaries (shadow zones), as incident wave energy was imposed only along the offshore
boundary. To prevent discontinuities in the SWAN bathymetry, the part of the domain not covered by the MOHID domain was updated with the depths of the MOHID cross-shore boundaries.

The open boundary condition was defined as a null gradient for the sediment concentrations in the water column, as well as for the sediment mass evolution at the bottom column (or, equivalently, for bathymetry). A null gradient condition was also imposed at the open boundaries for the normal and tangential current velocities. The radiation condition of Blumberg and
Kantha (1985) was imposed for water level, Eq. (3), assuming a passive condition at the cross-shore boundaries ($T_d =$





$1e^{32}$s) and an active condition at the offshore boundary ($T_d = 1e^{-12}$s). Thus, the water level was imposed at the offshore boundary as equal to the initial condition (zero in this case) to maintain the average water level inside the model domain, otherwise it would continuously decrease. The effect of lateral friction in land boundaries was considered for a better representation of the flow around the groynes. The horizontal viscosity was set as 1.0 m$^2$s$^{-1}$.

The initial bathymetry was defined by considering an equilibrium profile of the form: $h = \beta y^{2/3}$ (Dean, 1991), where $\beta$ is a constant set to 0.12, and $y$ is the distance to the shoreline. The average slope is approximately 1:60 in the first 200 m from the shoreline, decreasing seaward. The maximum slope ($\alpha_{max}$) was defined as 1:50 for the bed slope corrections. The bathymetry evolution was allowed only after a warm-up period, considering a morphological acceleration factor of 365. This means that 1 day of simulation time is equivalent to 1 year of morphological changes. The wave forcing was updated in

MOHID, as the bathymetry and water levels were updated in SWAN, with a constant frequency of 5 min (or 30 h of morphological evolution).

### 4.1.1 Detached Breakwaters

Detached breakwaters generate sediment transport from the adjacent coast to the lee side of the structure, leading to the formation of a bulge or salient in the beach planform. Depending on geometrical features of the breakwater, wave climate

and sediment availability, the salient may become attached to the breakwater forming a tombolo. Based on the analysis of several existing breakwater projects, Dally and Pope (1986) found that a ratio ($r$) between breakwater's length and distance to the shoreline less than 0.5 prevents the development of a tombolo. In contrast, the development of a tombolo is assured if $r$ is larger than 1.5, assuming sufficient sediment supply. Taking these values into account, we tested the model response for the four different detached breakwater designs described in Table 1.

Models results for a near equilibrium planform of the shoreline agree with the analysis of Dally and Pope (1986), demonstrating the development of a salient for $r$ equal to 0.2, 0.5 and 1.0 (Fig. 1, Fig. 2, Fig. 3), which become attached to the breakwater forming a tombolo only for $r$ equal to 2.0 (Fig. 4). The obliquity of waves generates a longshore current and, consequently, longshore sediment transport in the nearshore zone. The shoreline tends to be parallel to the wave crests, creating asymmetric bulges. For $r$ equal to 0.5 and 1.0, longshore currents restricted the size of the salients, preventing the

connection with the breakwater. The shoreline advances more on the updrift side for larger values of $r$, trapping sediment from the littoral drift. On the other hand, the downdrift beach erosion enhances. When a tombolo is formed, the detached breakwater affects the shoreline similarly to a groyne.

### 4.1.2 Groynes

Groynes are applied to reduce the littoral drift in the surf zone, trapping sediment on the updrift side of the structure, which

may cause erosion problems on the downdrift side. Moreover, the longshore currents are forced to deviate into deeper water around groynes, causing sediment losses from nearshore to offshore. The morphological impacts of the groynes are a function of their length from the shoreline. Model results were assessed for two designs of groynes, with lengths of 100 m



and 200 m (Fig. 5 and Fig. 6). As expected, greater erosion occurs on the downdrift side for a longer groyne length, as more sediment from the littoral drift gets trapped on the updrift side. Furthermore, offshore sediment transport becomes intensified in the 200 m length groyne design, as the deviation of longshore currents is more important. In this case, the retrogradation of the shoreline is similar to the case of a detached breakwater in which a tombolo was formed.

## 4.2 Sandbars Formation

In this test case, we verified the model capacity to generate sandbars in a 3D approach. The same domain and sand granulometry (0.2 mm) as in the previous 2DH test cases was considered, but ignoring defence structures. Two wave heights (1.5 m and 1.0 m) were defined in sequence along the offshore boundary during periods of 45 days of morphological evolution (3 h of simulation time with a morphological acceleration factor of 365). The peak wave period and peak wave direction were maintained constant (8 s and 15º). For the 3D case, a larger maximum slope was defined as 1:10 for the bed slope corrections, considering that the seaward sediment transport due to undertow currents can now be represented. The maximum slope in 3D simulations is useful to represent the sand motion induced by excessively steep slopes.

The water column was divided into five sigma layers and a simple exponential approach was followed to consider the vertical variation of the wave-induced forces: an exponential decrease of the radiation stress is imposed from the surface to the bottom, following the same shape as the profile of the orbital velocities, provided by the linear wave theory. The vertical radiation stress profile is designed to conserve the vertically integrated flux of momentum, which remains equal to the barotropic flux given by SWAN. The idea here is only to provide an approximate representation of the vertical distribution of wave momentum, in order to generate a general undertow pattern. Thus, the corresponding results should be considered as a first qualitative evaluation of the effect of such an undertow in our morphological module, the latter being our focus here. It is left for further work to use a more advanced formulation of 3D wave-current interactions for more quantitative investigations. The k-ε turbulence-closure model was used to compute the vertical viscosity, with the MOHID default parameterization, whereas the horizontal viscosity was set as $1.0 \text{ m}^2\text{s}^{-1}$.

The open boundary condition for sediment concentrations in the water column and sediment mass evolution at the bottom column was defined as a null gradient, as well as the boundary conditions for normal and tangential velocities, as in the previous 2DH test cases. Considering that defence structures were ignored in this test case, the bathymetry evolution was expected to be nearly uniform along the beach. Thus, a cyclic boundary condition was imposed at the cross-shore boundaries together with a Flather radiation condition at the offshore boundary, Eq. (4).

The model was capable of representing an undertow pattern and associated sediment transport that induces the formation of longshore sandbars (Fig. 7). A longitudinal current is presented in the surf zone, similarly to that observed in 2DH. However, a cross-shore velocity component is now represented by the 3D model. Inside the surf zone, this component is shoreward near the surface and seaward near the bottom. As expected, the cross-shore component has opposite directions before and after the breaking zone. The sandbar migrated seaward, changing the location of the breaking zone until wave heights decreased to 1.0 m after 45 days. At this time, waves were able to propagate further without breaking, creating a new



sandbar nearer to the coast. Finally, the results demonstrate the model's potential to represent the formation of multiple sandbars, which are observed in many places (e.g., Dolan and Dean, 1985; Ruessink et al., 2009).

## 4.3 Costa da Caparica

The hydrodynamics and sediment transport in the southern coast of the Tagus Estuary mouth (Costa da Caparica) are
evaluated under extreme wave conditions by coupling the MOHID modelling system and the SWAN wave model. A significant coastline retreat was observed in the Costa da Caparica in the last century. Defence structures (groynes) were built around the 1960's to reduce coastal erosion, resulting in some stability until the 2000/2001 winter when this issue started to receive more attention from the local authorities (Veloso-Gomes et al., 2009). The importance of the problem has augmented due to urbanisation and tourism development. The location near to the Tagus Estuary inlet increases the
complexity of sediment dynamics in this zone.

A downscaling approach was followed to provide appropriate boundary conditions for the Costa da Caparica model, considering previous results of the wave and hydrodynamic modelling system for the Portuguese Coast developed by the MARETEC research group (http://forecast.maretec.org). A new domain was created between the capes Raso and Espichel with 100 m x 100 m of grid resolution to propagate the waves until the Tagus Estuary mouth. The bathymetries of the father
hydrodynamic model and wave model are presented in Fig. 8, showing the domain of the Costa da Caparica model. The hydrodynamic boundary conditions (water level, current velocities, salinity, and temperature) for the Costa da Caparica model were provided by the three-dimensional baroclinic model for the Tagus Estuary (see Franz et al., 2014a, 2014b), through the application of the relaxation scheme together with the Flather radiation condition. The wave model for the Portuguese Coast was validated previously considering the data of the Port of Lisbon wave buoy, among others buoys
located along the Portuguese Coast (see Franz et al., 2014c).

The Tagus estuary is classified as mesotidal, with an average tidal height of 2.0 m in the mouth (Lemos, 1972). The tide is the main mechanism forcing the flow in the estuary, determining current directions and water level variations (Franz et al., 2014d). The maximum velocities reach up 2 m s$^{-1}$ in the estuary mouth. The Tagus River is the estuary's main freshwater source, with an annual average flow of about 300 m$^3$ s$^{-1}$. The estuary stratification is strongly related to the river flow and
tidal cycle. The residual currents pattern in the Tagus Estuary mouth is characterised by a jet in the inlet channel and two adjacent vortexes, originating a residual recirculation into the inlet estuary direction in front of the Costa da Caparica (Fig. 9).

The effects of the waves on the currents and sediment transport were investigated during a high energy event in the winter of 2013/2014 caused by Hercules storm. The wave conditions for the period of study in the location of the Port of Lisbon wave
buoy reached wave heights higher than 7 m and wave periods up to 20 s (Fig. 10). A variable grid resolution was defined for the Costa da Caparica domain, ranging from 50 m to 10 m near the coast. The water column was divided into ten layers, including five layers in the first metre above the bottom with fixed thickness ranging from 0.1 m to 0.3 m, and five sigma layers on top. The vertical viscosity was computed through the k-ε turbulence-closure model and the horizontal viscosity was



set as 5.0 m$^2$s$^{-1}$. The bathymetry data for the coast of the Costa da Caparica was provided by the Portuguese Environment Agency (APA).

In a first scenario, the effects of the waves on the currents and sediment transport were neglected. Therefore, just the influence of hydrodynamic boundary conditions from the Tagus Estuary model was taken into account. The results of the Costa da Caparica model without the wave action demonstrate strong velocities up to 2 m s$^{-1}$ at the surface and 0.5 m s$^{-1}$ near the bed in the northern zone (Cova do Vapor) with opposite directions on the flood and ebb tides, whereas weak velocities are found in the remaining coast directed to the inlet estuary in both situations (Fig. 11). The strong tidal currents are deflected away from the shoreline by the long groyne with approximately 500 m length located at Cova do Vapor.

During the period of study, the waves propagated from offshore mainly with a west-northwest (WNW) direction (Fig. 10). The bathymetric features (Fig. 8) cause the modification of the wave propagation direction through refraction to a west-southwest (WSW) or southwest (SW) direction in the nearshore (Fig. 12). The effect of the currents and water level variations on wave propagation was also considered, with an update frequency of 1 h, the same frequency in which the wave forcing was updated in MOHID. The oblique angle of the waves' incidence generates a nearshore longitudinal current oriented to the estuary inlet (Fig. 13), reinforcing the velocities observed in the scenario in which the wave action was neglected. The velocity vectors have a shoreward component at the surface, whereas near the bottom a seaward component is observed, caused by the vertical variation of the wave-induced forces. Although the currents were intensified along the coast of the Costa da Caparica, a small reduction of velocities can be observed in the northern part of the model domain.

Based on few existing data in the literature, a uniform granulometry was assumed for the Costa da Caparica model with a diameter of 0.3 mm (Freire, 2006). A more representative set of granulometry data is necessary to better characterise the sediment distribution in the model domain, considering that grain-size sorting is expected to occur due to the complex hydrodynamics, wave variability and bathymetric heterogeneities. Thus, the results of sediment transport should be seen as a first qualitative assessment. Moreover, the morphological evolution was ignored at this stage.

The patterns of the bed load sediment transport for the scenarios without waves and under extreme wave conditions are presented in Fig. 14. Along the coast of the Costa da Caparica, the results of the bed load sediment transport are practically null when the wave action was disregarded, suggesting that the tidal currents are irrelevant for the sediment transport in this area. On the contrary, simulations including wave forcing show a very strong bed load sediment transport due to waves. However, the importance of the tidal currents for the sediment transport in the inlet of the Tagus Estuary is noticeable. The littoral drift caused by the waves is deflected seaward by the tidal currents during the ebb tide and by the longer groynes present near the estuary inlet.

## 5 Conclusions

The potential of a new modelling approach to simulate the impact of different designs of coastal defence structures was demonstrated in this paper. The coupling between the MOHID modelling system and the SWAN wave model can be useful



for engineering studies in order to evaluate the best solution to protect the coast against erosion. The speed-up of morphological changes, along with the multiprocessing architecture, allows for the modelling of bed evolution for long periods and for the study of several scenarios. A more efficient coupling method is currently being developed inside the MOHID code to further reduce the computational time. Moreover, the interface of the MOHID modelling system (MOHID

Studio) is being developed to make the coupling with SWAN straightforward for all users.

The potential for modelling the evolution of sandbars was also demonstrated in this paper. In the future, an up-to-date methodology can be applied to resolve the vertical variation of wave-induced forces, as well as wave-induced vertical mixing, based on, e.g., the Generalised Lagrangian Mean (GLM) theory implemented in MOHID code (see Delpey et al., 2014). The test cases showed in this paper are only a preliminary demonstration of model potential, which is thought to be

encouraging. The findings of this work confirmed the applicability of the MOHID modelling system to study sediment transport and morphological changes in coastal systems under the combined action of waves and currents.

The application of the described modelling methodology to a coastal zone located near the inlet of a mesotidal estuary with strong tidal currents allowed for an assessment of the hydrodynamics and sediment transport in situations of calm water conditions (no waves) and under extreme wave conditions. Although these initial results are just a qualitative assessment of

sediment transport, the applicability of the modelling methodology to complex cases was demonstrated. In the future, with a more representative set of data, quantitative studies could be performed, taking into account the morphological evolution. Furthermore, the methodology can be used to evaluate different designs of defence structures in order to propose a more efficient solution for the coastline retreat and intense erosion observed in the last years on the coast of the Costa da Caparica.

**Acknowledgements**

The authors are grateful to the Portuguese Environment Agency (APA) for providing the bathymetry data for the coast of the Costa da Caparica. The first author is financed by the Brazilian National Council for Scientific and Technological Development (CNPq) under the *Ciências Sem Fronteiras* program (research grant no. 237448/2012-2).

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



**Table 1 – Detached breakwaters considered for test case scenarios**

| Length | Distance | Ratio ($r$) | Beach response* |
|--------|----------|-------------|-----------------|
| 100 | 500 | 0.2 | Salient |
| 100 | 200 | 0.5 | Salient |
| 200 | 200 | 1.0 | Salient/tombolo |
| 200 | 100 | 2.0 | Tombolo |

\* Beach response according to Dally and Pope (1986)



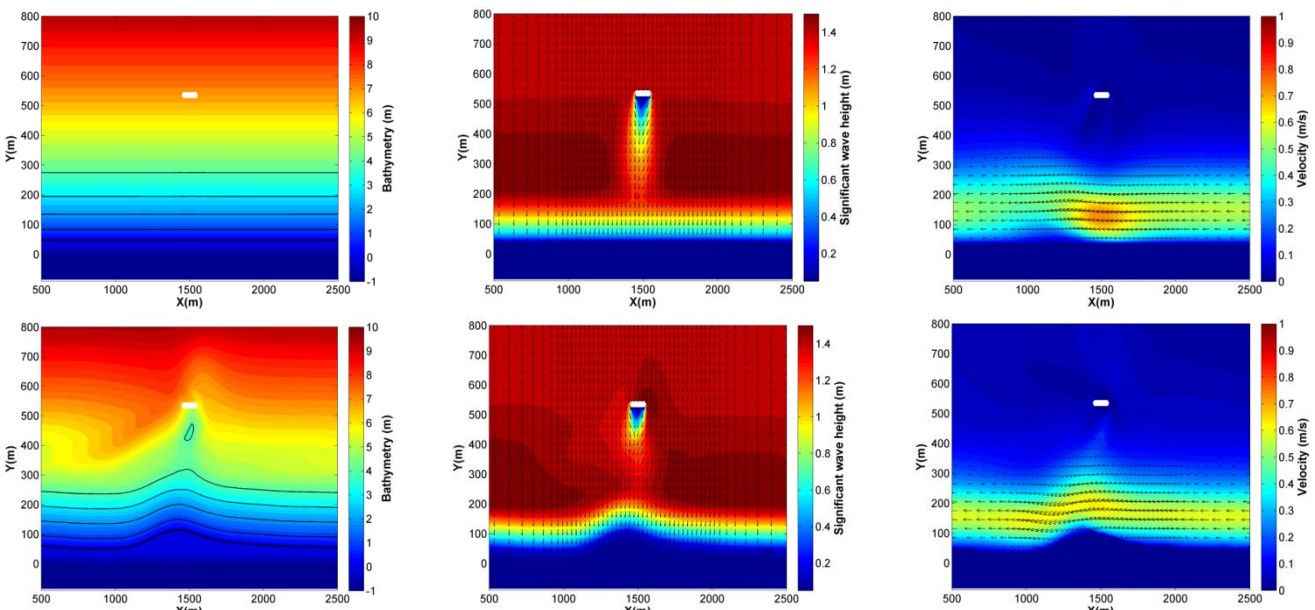

**Figure 1 – Model results for the breakwater's length to distance ratio of 0.2. Bathymetry (left), waves (middle) and currents (right) for the initial condition (above) and near equilibrium (below) after 9 years. The thicker isoline in the bathymetry represents the shoreline, whereas the remaining represent the 1 m, 2 m, 3 m and 4 m isobaths.**

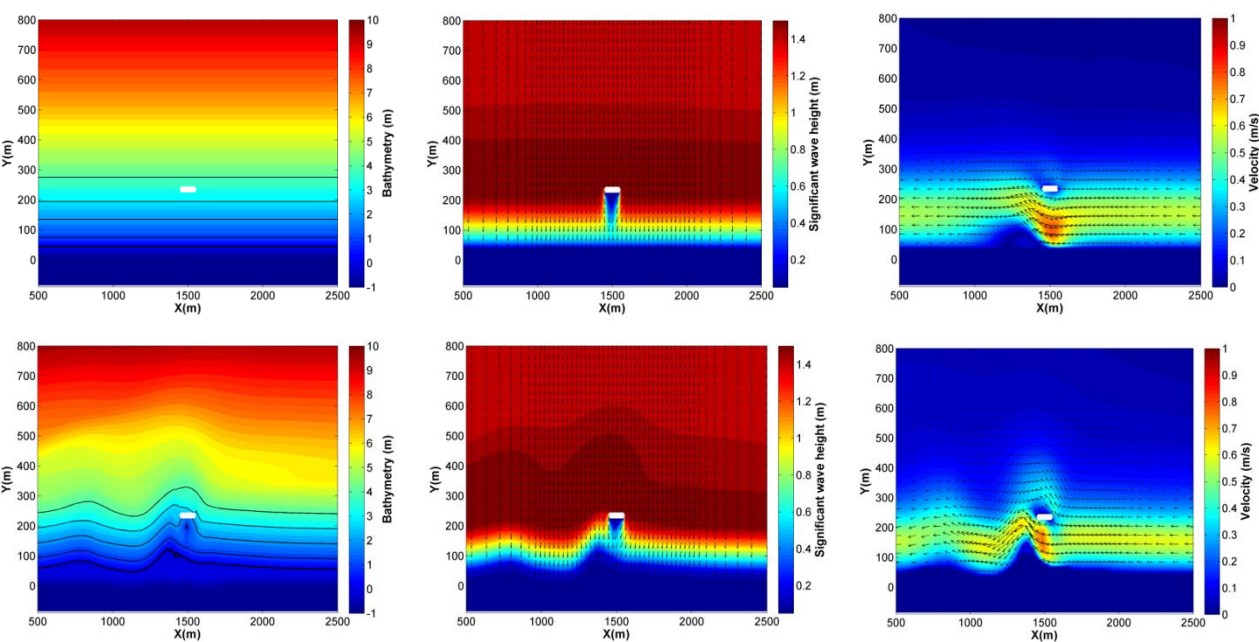

5     **Figure 2 – Model results for the breakwater's length to distance ratio of 0.5. Bathymetry (left), waves (middle) and currents (right) for the initial condition (above) and near equilibrium (below) after 18 years. The thicker isoline in the bathymetry represents the shoreline, whereas the others represent the 1 m, 2 m, 3 m and 4 m isobaths.**




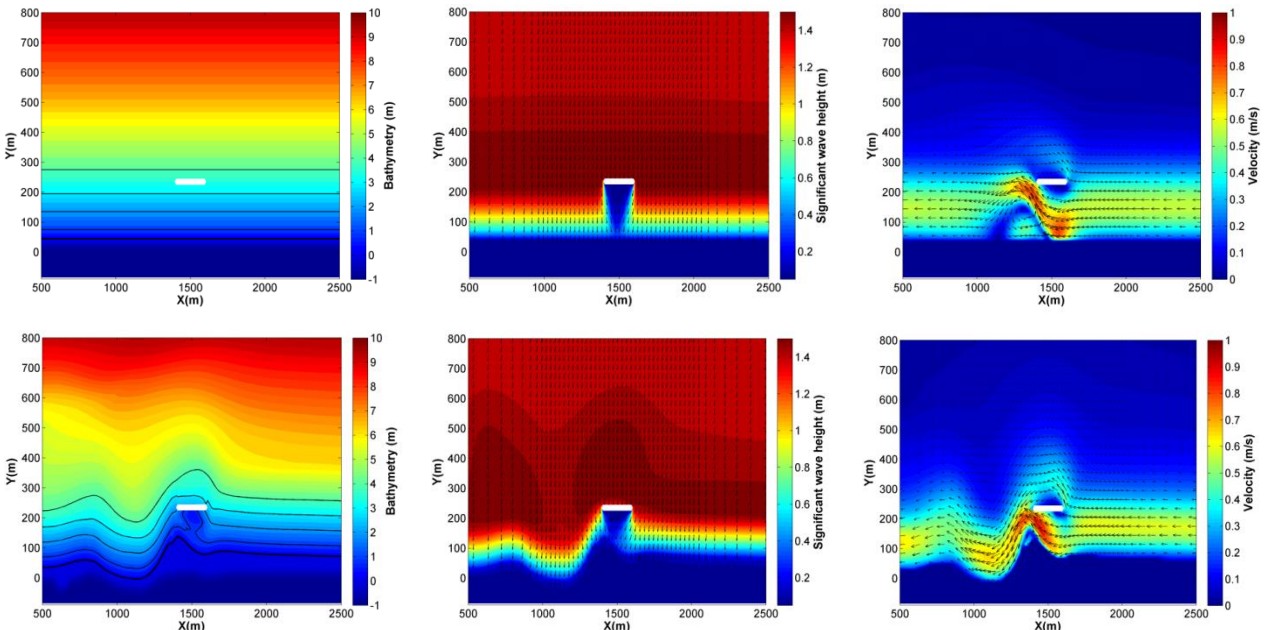

**Figure 3 – Model results for the breakwater's length to distance ratio of 1.0. Bathymetry (left), waves (middle) and currents (right) for the initial condition (above) and near equilibrium (below) after 27 years. The thicker isoline in the bathymetry represents the shoreline, whereas the others represent the 1 m, 2 m, 3 m and 4 m isobaths.**

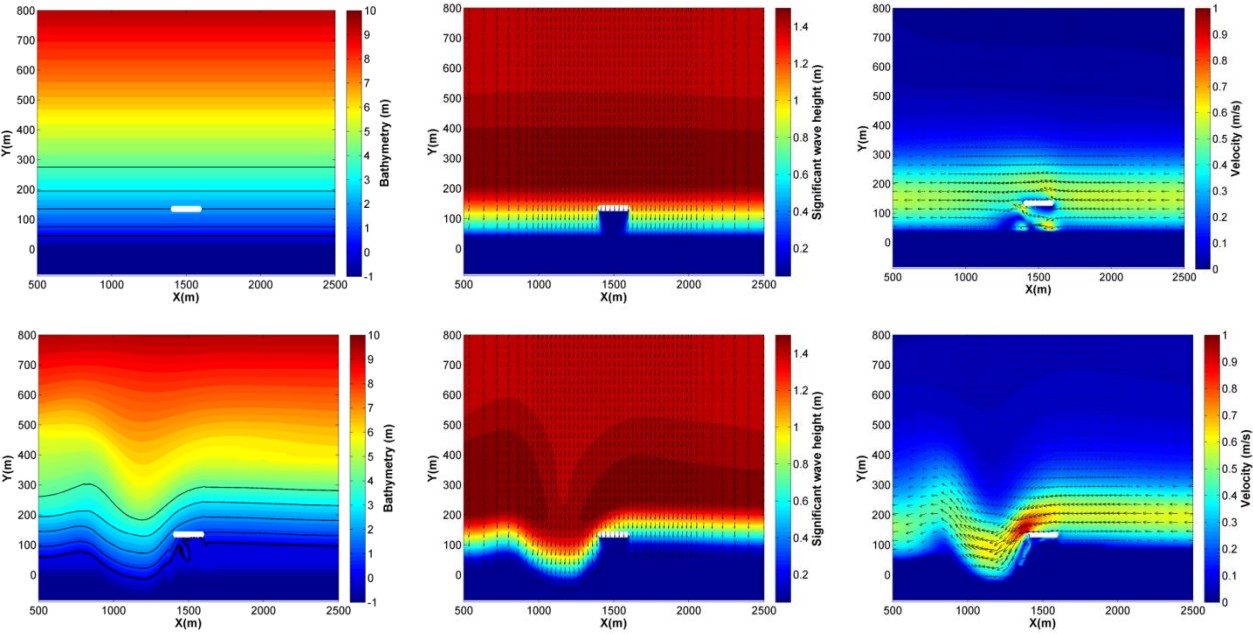

5    **Figure 4 - Model results for the breakwater's length to distance ratio of 2.0. Bathymetry (left), waves (middle) and currents (right) for the initial condition (above) and near equilibrium (below) after 18 years. The thicker isoline in the bathymetry represents the shoreline, whereas the others represent the 1 m, 2 m, 3 m and 4 m isobaths.**





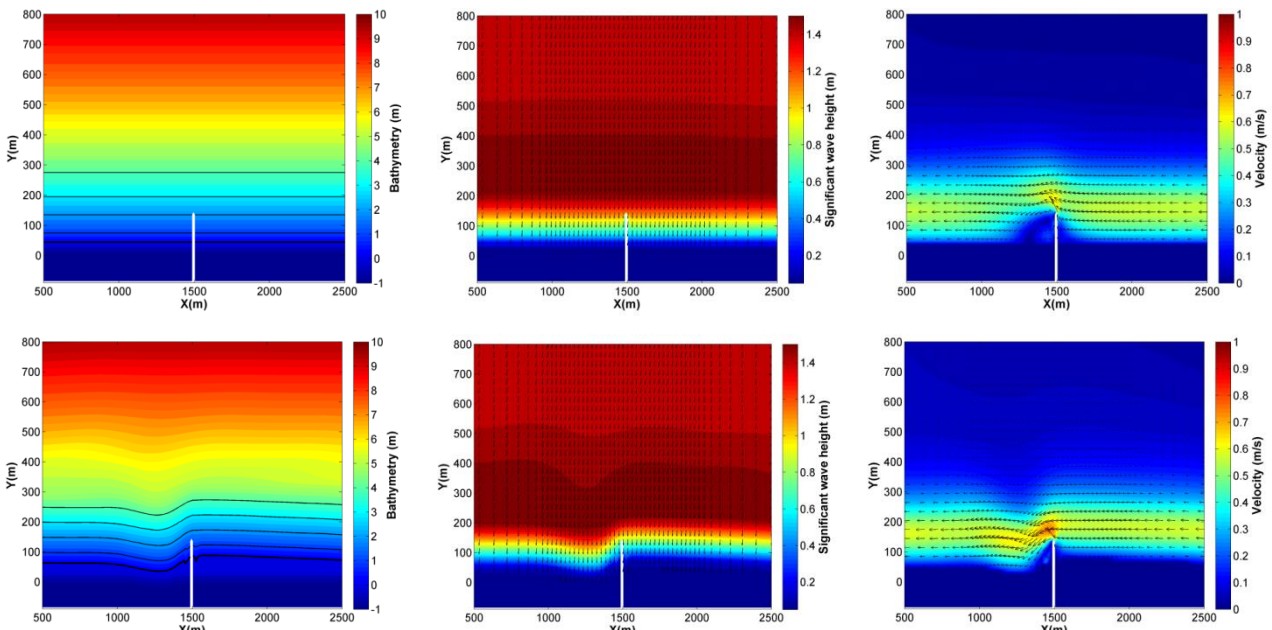

**Figure 5 - Groyne with a length of 100 m from the shoreline. Bathymetry (left), waves (middle) and currents (right) for the initial condition (above) and near equilibrium (below) after 9 years. The thicker isoline in the bathymetry represents the shoreline, whereas the others represent the 1 m, 2 m, 3 m and 4 m isobaths.**

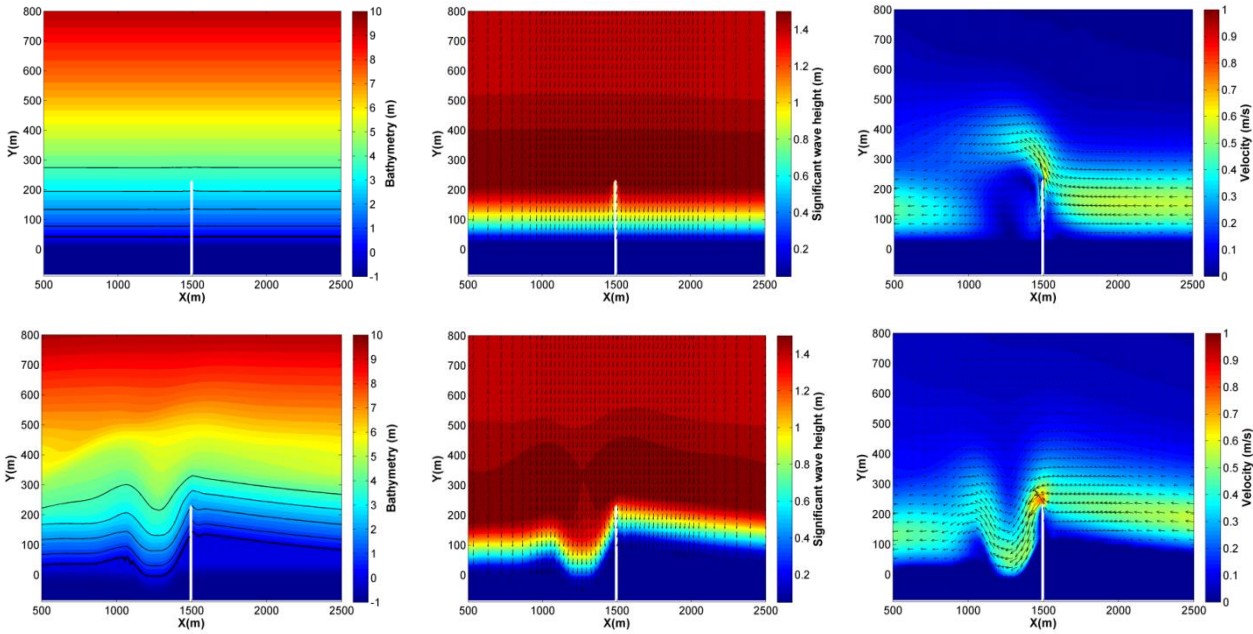

5 **Figure 6 – Groyne with a length of 200 m from the shoreline. Bathymetry (left), waves (middle) and currents (right) for the initial condition (above) and near equilibrium (below) after 9 years. The thicker isoline in the bathymetry represents the shoreline, whereas the others represent the 1 m, 2 m, 3 m and 4 m isobaths.**

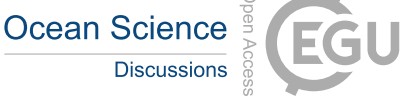



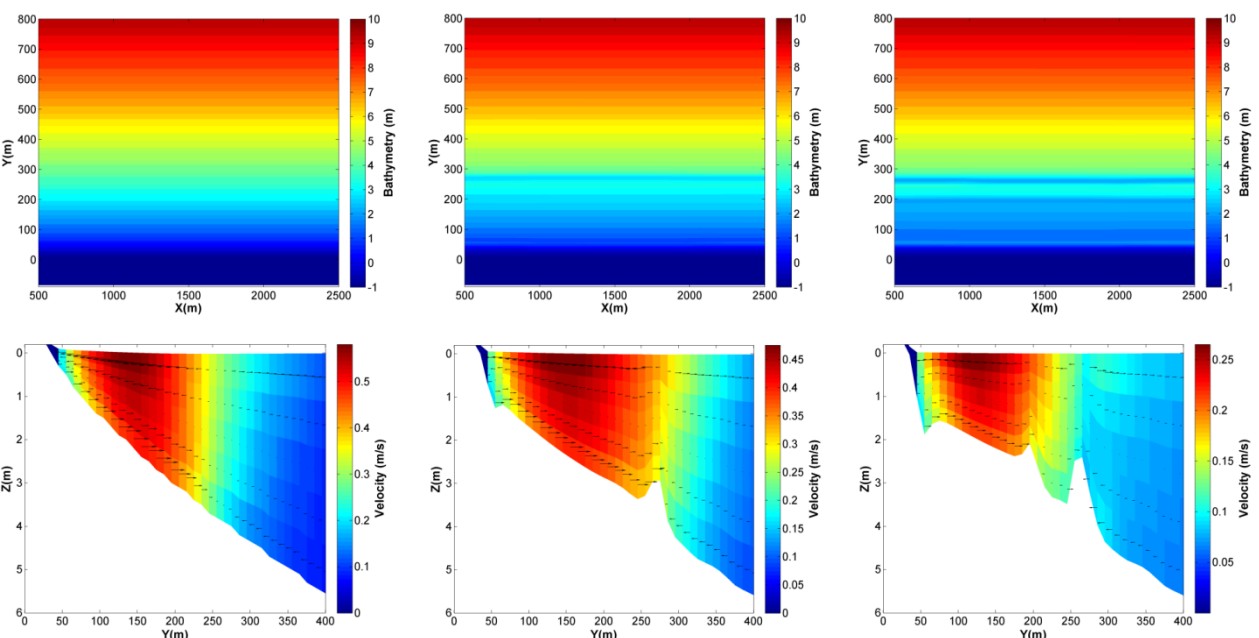

**Figure 7 – Simulated sandbars for different wave heights. The horizontal plane of the bathymetry (above) and a vertical cut with velocity modulus and vectors (below). Left: Initial condition. Middle: After 45 days of morphological evolution with 1.5 m wave height. Right: After additional 45 days of morphological evolution with 1.0 m wave height.**

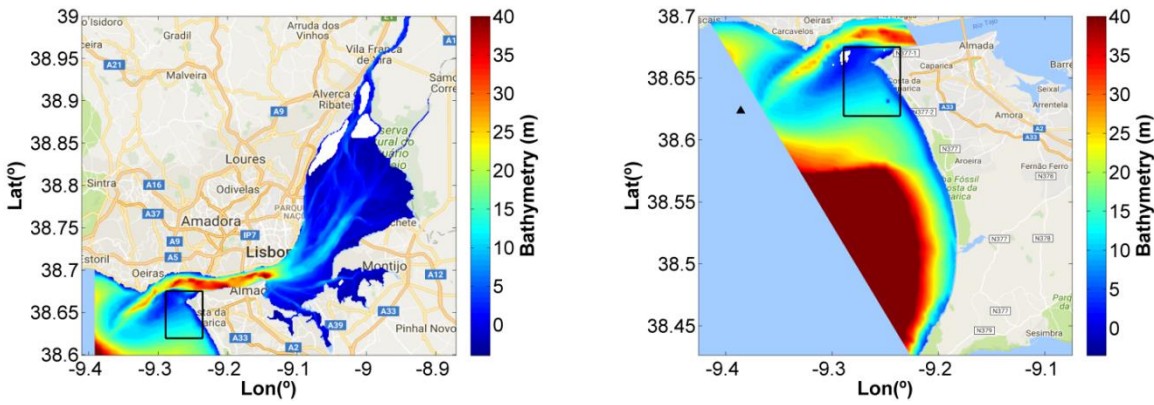

5  **Figure 8 – Bathymetries of the Tagus Estuary hydrodynamic model (left) and the wave model (right) created to propagate the waves until the Tagus Estuary mouth. The domain of the Costa da Caparica model is presented by the black rectangle and the location of the Port of Lisbon wave buoy is indicated by the black triangle.**



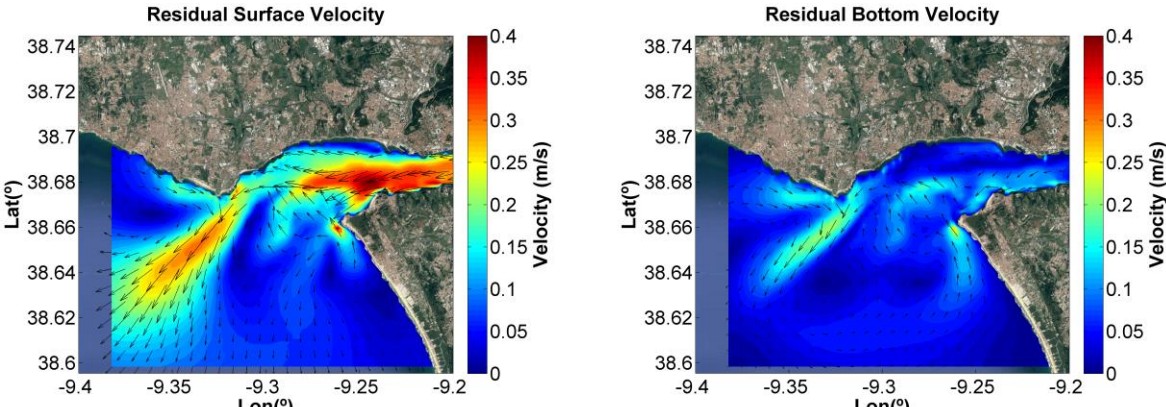

**Figure 9 – Residual velocity in the Tagus Estuary mouth at the surface (left) and near the bottom (right) obtained from the three-dimensional baroclinic model for the Tagus Estuary.**

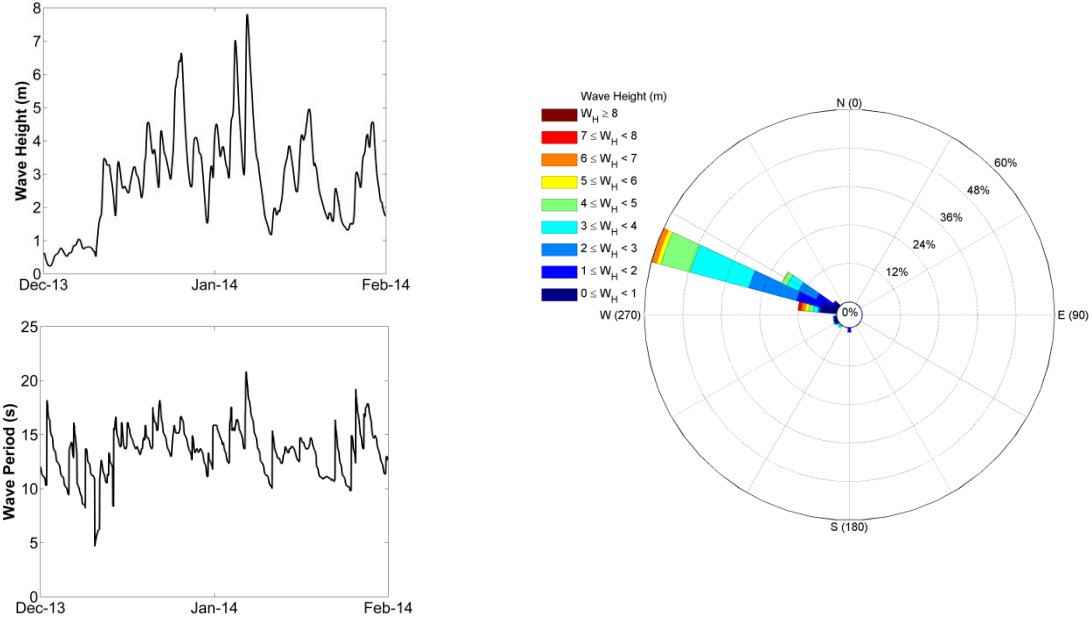

**Figure 10 - Wave conditions for the winter of 2013/2014 obtained from the Portuguese Coast wave model in the location of the Port of Lisbon wave buoy.**



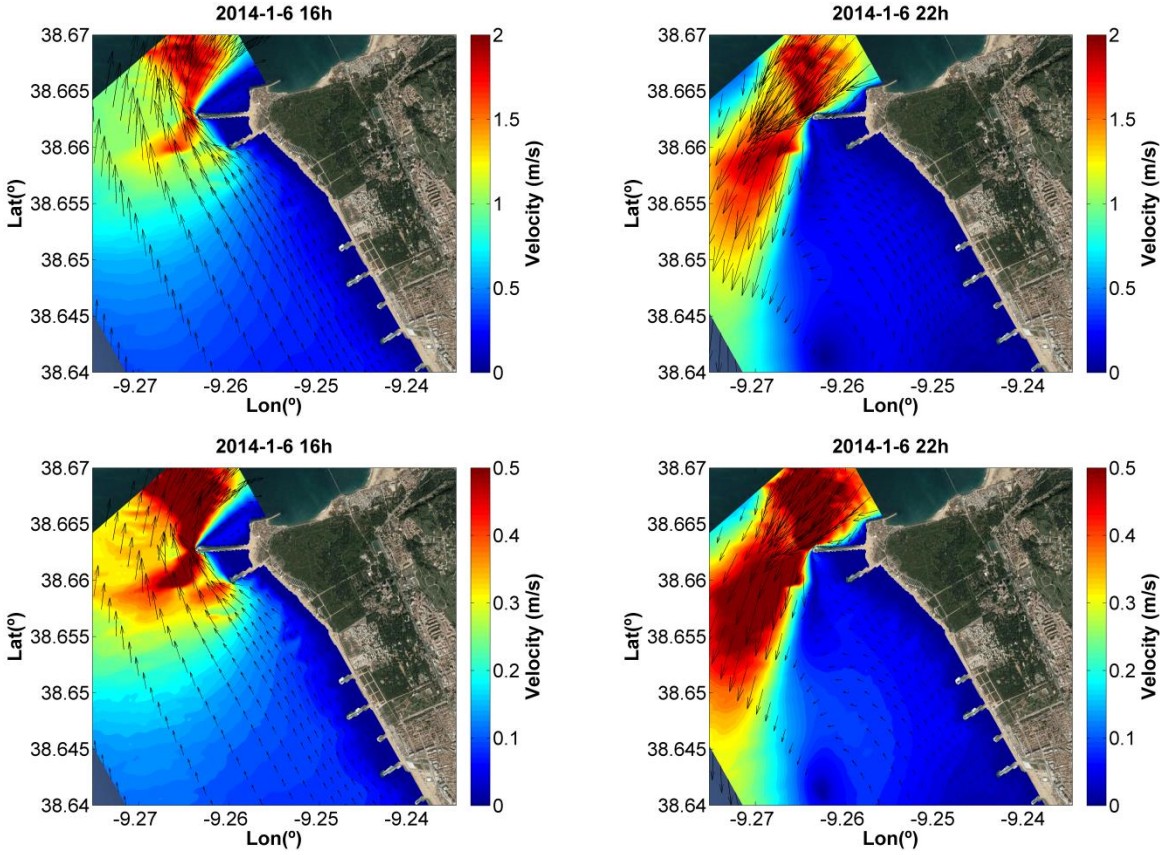

**Figure 11 – Velocity results of the Costa da Caparica model during flood (left) and ebb (right) tides at the surface (above) and near the bottom (below) without considering the wave's effect on the hydrodynamics.**

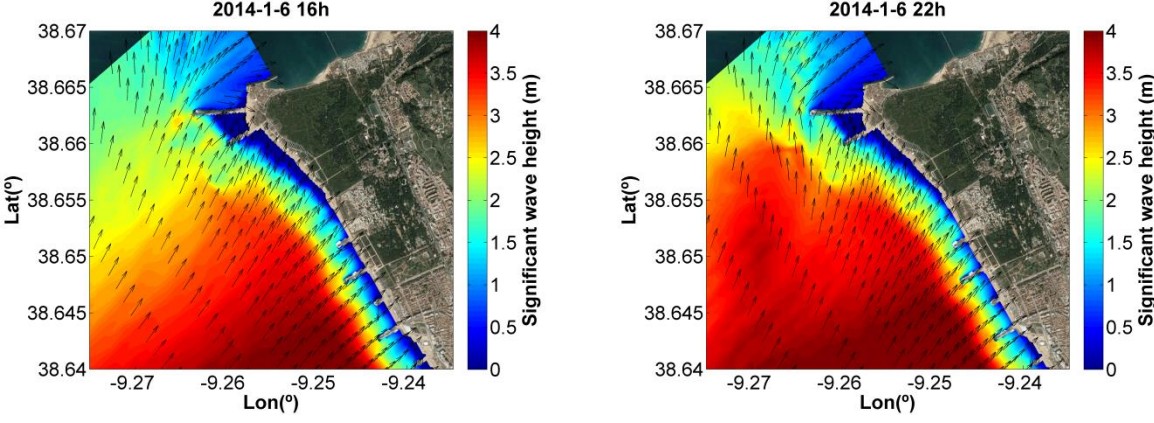

**Figure 12 – Wave results of the Costa da Caparica model.**





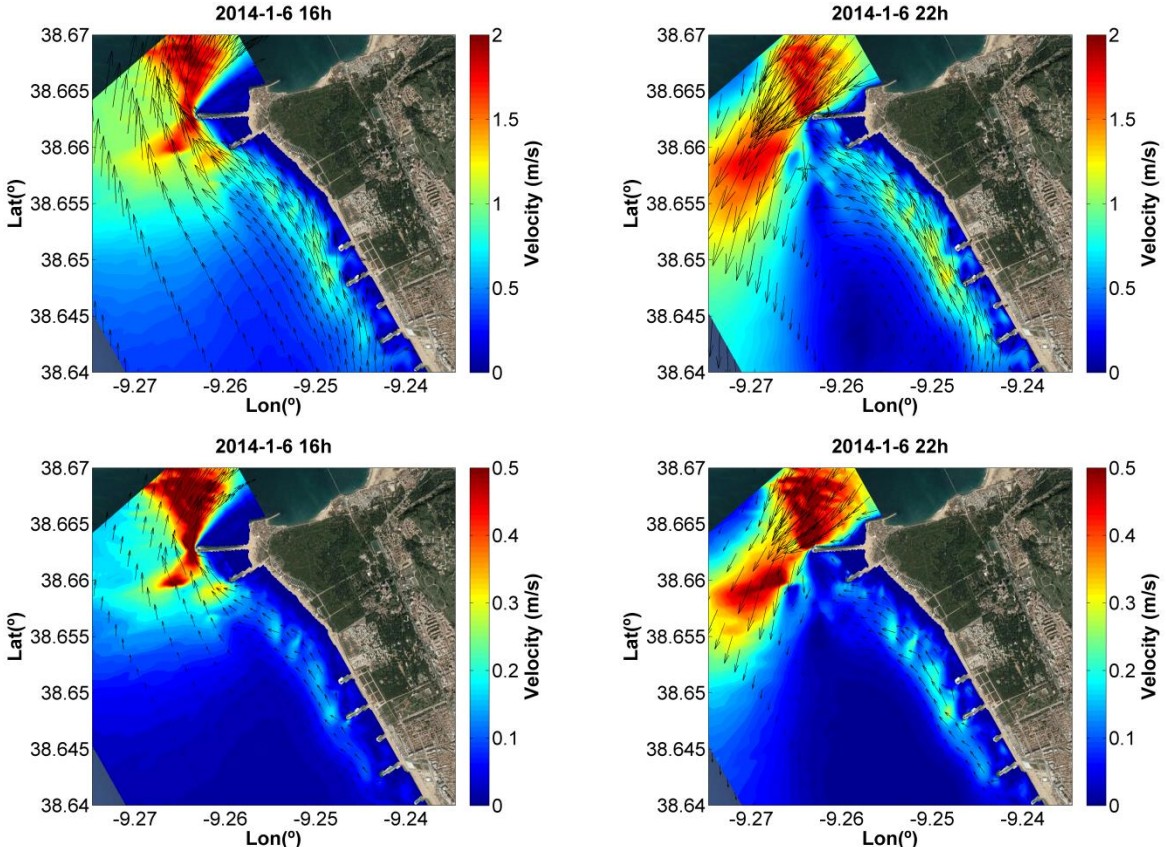

**Figure 13 - Velocity results of the Costa da Caparica model during flood (left) and ebb (right) tides at the surface (above) and near the bottom (below) considering the wave's effect on the hydrodynamics.**



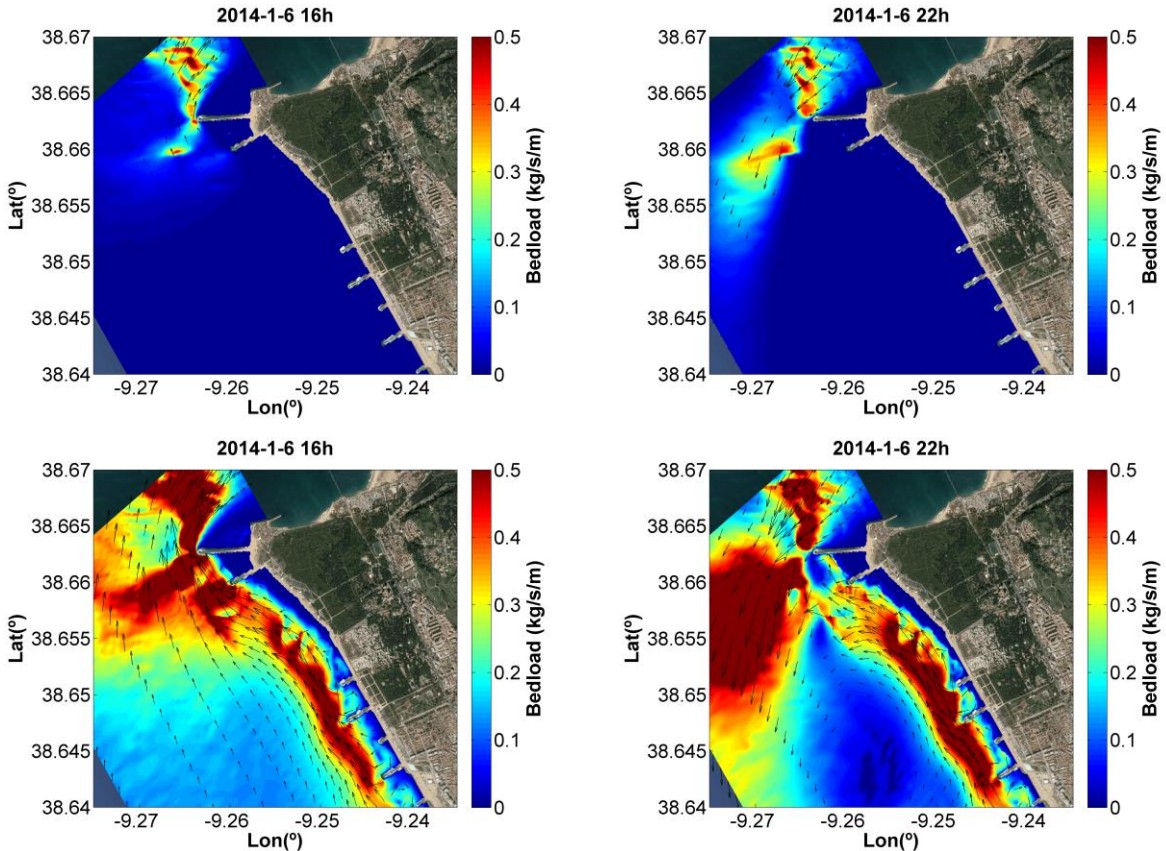

Figure 14 – Results of bed load sediment transport for the scenarios without waves (above) and under extreme wave conditions (below).

