# Peer review of "Modelling of sediment transport and morphological evolution under the combined action of waves and currents"

_Ocean Science, 2017_

## Referee Comment (RC1) · A. Fortunato (Referee) · 13 May 2017

General comments

This manuscript presents a new modeling system for coastal morphodynamics, coupling existing hydrodynamic (MOHID) and wave (SWAN) models, with a sediment transport and bottom evolution model. It is therefore in the scope of Ocean Science. The application of the modeling system is illustrated through several synthetic cases and a real beach/inlet case. Particular attention is given to the bed load transport. The results presented are realistic.

The manuscript is well organized and clear. The title reflects the content of the

manuscript. The detail of the model description is appropriate and the figures are clear. The test cases represent a wide range of conditions in which the model can be applied.

Overall, the manuscript is a good contribution to the scientific literature. However, some aspects lack a discussion. The paper – and the reader – would benefit from the insights that the authors gained during the development and application of the model. Some suggestions on how to enrich the paper are provided in the specific comments.

Specific comments

Introduction. The manuscript lacks a literature review on coastal area morphodynamic models, even though several models are mentioned. Such a review would allow the clarification of how the new model differs from existing ones and extends the state-of-the-art.

Page 2, line 32. The authors stress the use of a slope limiter as a solution to the deterioration of the results in long-term simulations. Yet, as is mentioned later (page 9, line 5), a similar filter was used before (Roelvink et al., 2009). Other authors use the same approach to improve numerical stability and to improve the simulation of tidal inlet migration (Nahon et al., 2012; Fortunato et al., 2014). This should be mentioned. The differences (if any) between the approach implemented by the authors and by Roelvink et al. (2009) should be mentioned and discussed. Also, since the authors consider this approach to be a significant contribution of the paper, they could show an example of a simulation without the slope limiter, in order to discuss its importance.

Page 2, line 34. "This approach may be more appropriate. . ." Explain why.

Section 3. More details should be provided about the coupling between the models. In particular, the way the information is exchanged between modules is critical for numerical efficiency and should be mentioned.

Section 3. An interesting aspect in the model is the solution of the transport equation

even in 2DH mode instead of the use of empirical formulae. This approach is not the most common, and has given poor results in the past (e.g., Galappatti and Vreugdenhil, 1985). A discussion on this issue would be very useful. For instance, it is unclear how the model deals with the vertical integration of the sediment fluxes. Are the velocity and the sediment concentration profiles assumed constant in the vertical? Are analytical profiles assumed? The approximations associated with this approach and the associated errors should be discussed. Similarly, the benefits should also be stressed, in particular the ability to represent the wash load, which cannot be represented by the equilibrium empirical formulae. Such discussions could be theoretical or based on a comparison between two simulations for the same test case, one in 2D and the other in 3D.

Page 4, line 20. In the description of the equations assumptions, incompressibility is missing. Also, it would seem more correct to call the governing equations "shallow water equations" than "Navier-Stokes equations".

Page 5, 1st paragraph. How is the turbulence associated to wave breaking taken into account?

Page 6, line 14. It is stated that lateral friction can be important, and it is mentioned later that it is taken into account in some tests. Please explain how is it computed in the model, or provide a reference.

Page 7, line 16. vH is defined as the current velocity. Yet here it is used to determine the wave bed shear stress. Please explain.

Page 7, equation 8. Considering that a spectral model is being used, what is the wave height Hw? Is it the significant wave height?

Page 11, line 11. "The maximum slope in 3D simulations is useful to represent the sand motion induced by excessively steep slopes." Are the slope effects included in the bedload formulation (e.g., as in Lesser et al., 2004)? If so, is the slope limiter used

to represent a physical process (explain which one) or to improve numerical stability?

Figure 10. This figure could be enriched by showing the forcing model's results, i.e, the time series that are actually being used in the simulation.

Figure 12. Two different time steps are shown. If the two are required, then a discussion of the differences should be useful. Otherwise, I suggest eliminating one of them.

Page 14, line 8. "A more efficient coupling method is currently being developed inside the MOHID code to further reduce the computational time." It would be useful to provide some indication on the computational performance of the model in its present stage.

Technical corrections

Page 2, line 7: "bi-" should be "two-"

Page 2, line 11: "for a hydrodynamic" should be "to a hydrodynamic". Similarly in line 12, "for a wave model" should be corrected.

Page 3, line 27: "generates" should be "generate"

Page 10, line 9: Dean's equation requires units.

Page 10, line 30: "enhances" should be "increases" or "grows"

Figs. 1-6. Better resolution is required. It is difficult to see the arrows' heads.

Page 11, line 6. "longer groyne length": remove "length"

Page 11, line 22. "which remains equal to the barotropic flux given by SWAN". By "barotropic" do the authors mean "depth-averaged"? I do not understand the use of the word "barotropic" in this context.

Page 12, line 20. Capes Raso and Espichel are mentioned in the text. They should be indicated in figure 8. Same for Cova do Vapor and other place names that I may have missed.

Page 13, line 7. "Caparica was" should be "Caparica were".

Page 13, line 18. "as the waves propagate with higher velocities". Replace "velocities" by "celerity".

Page 13, line 27. The reference Freire (2006) should probably be Freire et al. (2006). Also, this reference seems incomplete.

Page 17, line 21. "Saville, T.: Experimental determination of wave set-up, 1961." This reference is incomplete.

References

Fortunato, André B; Nahon, Alphonse; Dodet, Guillaume; Rita Pires, A; Conceição Freitas, M; Bruneau, Nicolas; Azevedo, Alberto; Bertin, Xavier; Benevides, Pedro; Andrade, César; Oliveira, Anabela. 2014. Morphological evolution of an ephemeral tidal inlet from opening to closure: The Albufeira inlet, Portugal, Continental Shelf Research 73, -: 49 - 63.

Galappatti, G, C.B. Vreugdenhil (1985). A depth-integrated model for suspended sediment transport, Journal of Hydraulic Research, 23/4: 359-377.

Lesser, G.R., Roelvink, J.A., van Kester, J.A.T.M., Stelling, G.S., 2004. Development and validation of a three-dimensional morphological model. Coastal Engineering 51, 883–915.

Nahon, Alphonse; Bertin, Xavier; Fortunato, André B; Oliveira, Anabela. 2012. Process-based 2DH morphodynamic modeling of tidal inlets: A comparison with empirical classifications and theories, Marine Geology 291294, 1: 1 - 11.

Roelvink, D., Reniers, A., Van Dongeren, A., de Vries, J. v. T., McCall, R., and Lescinski, J.: Modelling storm impacts on 15 beaches, dunes and barrier islands, Coastal engineering, 56, 1133-1152, 2009.

---

## Referee Comment (RC2) · Anonymous Referee #2 · 2 Jun 2017

An annotated version of the manuscript is being attached, and those comments are not going to be repeated here.

The paper is well written, the title is clear and reflects the paper's content.

I missed references of some important publications in the field of nearshore dynamics such as I. A. Svendsen's book by world scientific "Introduction to Nearshore Hydrodynamics" wherein many papers on the subject can be found by the author and several associates (Putrevu, etc). They have even made available a model named SHORE-CIRC.

The acceleration factor of 365 for sediment transport used in the paper appears to me

to be too high. A little discussion on the effect of this factor on the simulation would be nice. I also wonder if such high factor wouldn't limit the application of the model in cases where the wave spectrum varies on a daily or weekly basis.

The coupling of Swam and MOHID is a huge task, but it was not clear to me whether or not it was necessary to iterate solutions between the 2 models so that wave affected currents and vice-versa.

The inclusion of the wave module into MOHID clearly made a huge difference, but it was unclear to me the effectiveness for the maximum bottoms slope criterion. Perhaps one run with that criterion relaxed and comparison with what was done would be interesting.

Please also note the supplement to this comment:
http://www.ocean-sci-discuss.net/os-2017-8/os-2017-8-RC2-supplement.pdf

[Figure]

**Supplement:**

[revised manuscript text omitted]

---

## Author Comment (AC2) · 8 Jun 2017

We are grateful to the referee for his comments and contributes to the manuscript improvement. The manuscript was thoroughly revised to address the referee comments. Our answers to the main questions raised may be found below. Other corrections were performed directly on the manuscript (new manuscript attached as supplement file).

The acceleration factor of 365 allowed us to simulate many years of morphological evolution of a schematic beach with constant wave conditions in a feasible computational time. The bathymetry results reached an equilibrium condition, demonstrating the model stability. A discussion on the effect of the acceleration factor in model results

was presented previously in Franz et al. (2017). The acceleration factor is user-defined, depending on the variability of forcing conditions and speed of morphological changes. The morphological changes should not be speeded up (acceleration factor = 1) to simulate extreme events that occur within few days, as for the case presented for the Costa da Caparica.

The wave-induced forces computed by the SWAN wave model are provided to the MOHID hydrodynamic model in order to simulate wave-related phenomena, such as wave-induced currents. On the other hand, the MOHID hydrodynamic model can return water levels and currents to the wave model. The water level variation caused, for instance, by the tidal motion changes the breaker zone and shoreline position, affecting waves and sediment transport. The morphological evolution also modifies the currents and waves. Due to the interdependence of the physical processes involved, SWAN and MOHID models must be coupled, and the different fields computed by SWAN (e.g., wave-induced force) must be updated in MOHID, as well as the different fields computed by MOHID (e.g., water level, bathymetry) must be updated in SWAN, with an adequate frequency for each application.

In order to demonstrate the importance of the maximum bottom slope criterion, bathymetry results of a simulation without considering bed slope corrections was added to the manuscript (Fig. 1), as suggested by the referee. Thus, the effectiveness of this method is demonstrated in a clearer manner.

[Figure]

[Figure]

**Supplement:**

[revised manuscript text omitted]

---

## Author Response (AR1)

General comments

This manuscript presents a new modeling system for coastal morphodynamics, coupling existing hydrodynamic (MOHID) and wave (SWAN) models, with a sediment transport and bottom evolution model. It is therefore in the scope of Ocean Science. The application of the modeling system is illustrated through several synthetic cases and a real beach/inlet case. Particular attention is given to the bed load transport. The results presented are realistic.

The manuscript is well organized and clear. The title reflects the content of the

manuscript. The detail of the model description is appropriate and the figures are clear. The test cases represent a wide range of conditions in which the model can be applied.

Overall, the manuscript is a good contribution to the scientific literature. However, some aspects lack a discussion. The paper – and the reader – would benefit from the insights that the authors gained during the development and application of the model. Some suggestions on how to enrich the paper are provided in the specific comments.

Specific comments

Introduction. The manuscript lacks a literature review on coastal area morphodynamic models, even though several models are mentioned. Such a review would allow the clarification of how the new model differs from existing ones and extends the state-of-the-art.

Page 2, line 32. The authors stress the use of a slope limiter as a solution to the deterioration of the results in long-term simulations. Yet, as is mentioned later (page 9, line 5), a similar filter was used before (Roelvink et al., 2009). Other authors use the same approach to improve numerical stability and to improve the simulation of tidal inlet migration (Nahon et al., 2012; Fortunato et al., 2014). This should be mentioned. The differences (if any) between the approach implemented by the authors and by Roelvink et al. (2009) should be mentioned and discussed. Also, since the authors consider this approach to be a significant contribution of the paper, they could show an example of a simulation without the slope limiter, in order to discuss its importance.

Page 2, line 34. "This approach may be more appropriate. . ." Explain why.

Section 3. More details should be provided about the coupling between the models. In particular, the way the information is exchanged between modules is critical for numerical efficiency and should be mentioned.

Section 3. An interesting aspect in the model is the solution of the transport equation

even in 2DH mode instead of the use of empirical formulae. This approach is not the most common, and has given poor results in the past (e.g., Galappatti and Vreugdenhil, 1985). A discussion on this issue would be very useful. For instance, it is unclear how the model deals with the vertical integration of the sediment fluxes. Are the velocity and the sediment concentration profiles assumed constant in the vertical? Are analytical profiles assumed? The approximations associated with this approach and the associated errors should be discussed. Similarly, the benefits should also be stressed, in particular the ability to represent the wash load, which cannot be represented by the equilibrium empirical formulae. Such discussions could be theoretical or based on a comparison between two simulations for the same test case, one in 2D and the other in 3D.

Page 4, line 20. In the description of the equations assumptions, incompressibility is missing. Also, it would seem more correct to call the governing equations "shallow water equations" than "Navier-Stokes equations".

Page 5, 1st paragraph. How is the turbulence associated to wave breaking taken into account?

Page 6, line 14. It is stated that lateral friction can be important, and it is mentioned later that it is taken into account in some tests. Please explain how is it computed in the model, or provide a reference.

Page 7, line 16. vH is defined as the current velocity. Yet here it is used to determine the wave bed shear stress. Please explain.

Page 7, equation 8. Considering that a spectral model is being used, what is the wave height Hw? Is it the significant wave height?

Page 11, line 11. "The maximum slope in 3D simulations is useful to represent the sand motion induced by excessively steep slopes." Are the slope effects included in the bedload formulation (e.g., as in Lesser et al., 2004)? If so, is the slope limiter used

[Figure]

to represent a physical process (explain which one) or to improve numerical stability?

Figure 10. This figure could be enriched by showing the forcing model's results, i.e, the time series that are actually being used in the simulation.

Figure 12. Two different time steps are shown. If the two are required, then a discussion of the differences should be useful. Otherwise, I suggest eliminating one of them.

Page 14, line 8. "A more efficient coupling method is currently being developed inside the MOHID code to further reduce the computational time." It would be useful to provide some indication on the computational performance of the model in its present stage.

Technical corrections

Page 2, line 7: "bi-" should be "two-"

Page 2, line 11: "for a hydrodynamic" should be "to a hydrodynamic". Similarly in line 12, "for a wave model" should be corrected.

Page 3, line 27: "generates" should be "generate"

Page 10, line 9: Dean's equation requires units.

Page 10, line 30: "enhances" should be "increases" or "grows"

Figs. 1-6. Better resolution is required. It is difficult to see the arrows' heads.

Page 11, line 6. "longer groyne length": remove "length"

Page 11, line 22. "which remains equal to the barotropic flux given by SWAN". By "barotropic" do the authors mean "depth-averaged"? I do not understand the use of the word "barotropic" in this context.

Page 12, line 20. Capes Raso and Espichel are mentioned in the text. They should be indicated in figure 8. Same for Cova do Vapor and other place names that I may have missed.

[Figure]

Page 13, line 7. "Caparica was" should be "Caparica were".

Page 13, line 18. "as the waves propagate with higher velocities". Replace "velocities" by "celerity".

Page 13, line 27. The reference Freire (2006) should probably be Freire et al. (2006). Also, this reference seems incomplete.

Page 17, line 21. "Saville, T.: Experimental determination of wave set-up, 1961." This reference is incomplete.

[Figure]

An annotated version of the manuscript is being attached, and those comments are not going to be repeated here.

The paper is well written, the title is clear and reflects the paper's content.

I missed references of some important publications in the field of nearshore dynamics such as I. A. Svendsen's book by world scientific "Introduction to Nearshore Hydrodynamics" wherein many papers on the subject can be found by the author and several associates (Putrevu, etc). They have even made available a model named SHORE-CIRC.

The acceleration factor of 365 for sediment transport used in the paper appears to me

to be too high. A little discussion on the effect of this factor on the simulation would be nice. I also wonder if such high factor wouldn't limit the application of the model in cases where the wave spectrum varies on a daily or weekly basis.

The coupling of Swam and MOHID is a huge task, but it was not clear to me whether or not it was necessary to iterate solutions between the 2 models so that wave affected currents and vice-versa.

The inclusion of the wave module into MOHID clearly made a huge difference, but it was unclear to me the effectiveness for the maximum bottoms slope criterion. Perhaps one run with that criterion relaxed and comparison with what was done would be interesting.

Please also note the supplement to this comment:
http://www.ocean-sci-discuss.net/os-2017-8/os-2017-8-RC2-supplement.pdf

—————————————————

[Figure]

Ocean Sci. Discuss.,
https://doi.org/10.5194/os-2017-8-AC1, 2017

[Figure]

We are grateful to Dr. Fortunato for his comments and contributes to the manuscript improvement. The manuscript was thoroughly revised to address the referee comments. Our answers to the main questions raised may be found below. Other smaller corrections were performed directly on the manuscript (new manuscript attached as supplement file).

In this manuscript, we introduced an overview of some publications that were important to this work and could guide the reader to a better understanding of the main topics discussed. A complete literature review was out of the scope of this manuscript. A

recent publication that gives more details about the model development and differences to other models was cited in the manuscript (see Franz et al., 2017).

Roelvink et al. (2009) used a slope delimiter to consider the erosion of dunes on beaches (avalanching mechanism). In our work, we implemented a similar approach to account for the neglected forces in 2DH models (e.g., undertow), allowing sediment transport seaward and avoiding excessive bed slopes (this is explained in section 3.3). The method was implemented in the MOHID code in terms of mass evolution, in order to be applied in the future considering multiple sediment fractions. This approach also permits to consider the shoreline evolution. The fact that other authors have used the same approach for simulating tidal inlet migration was unknown by the authors. The references of Nahon et al. (2012) and Fortunato et al. (2014) were added to the manuscript. In order to demonstrate the importance of the bed slope correction, bathymetry results of a simulation without considering this mechanism was added to the manuscript (Fig. 1), as suggested by the referee.

The approach adopted in this work, by defining a maximum slope that when surpassed generates sediment transport in the downslope direction, affects only individual cells instead of the entire beach profile. Other models consider the extrapolation of the erosion and deposition fluxes over the entire beach profile. In this case, the bathymetry and shoreline position are updated by defining an invariant equilibrium profile. Thus, the approach adopted in this work is more appropriate to consider the effect of grain-size sorting along the beach profile.

The coupling between the MOHID modelling system and the SWAN wave model was performed through tools developed in the Fortran language in order to convert the results to the appropriate format (section 3.4). This means that the coupling between models was performed by files transferring. An external tool was developed in Python language to automatically manage the runs of the tools and models. At this time, we have focused on model results instead of numerical efficiency, which is a suggestion for future work. However, considering the domain decomposition parallelization approach

implemented using MPI directives and the morphological acceleration factor, the computational time required to simulate the presented test cases was feasible through the use of a regular computer with 6 cores.

Explicitly resolving the advection-diffusion equation for suspended sediment leads to more realistic transport results and smoother bed evolution, as the suspended load is not in equilibrium with the instantaneous bed shear stresses in unsteady flows. The net upward flux of suspended sand depends on the equilibrium concentration near the bottom, estimated by empirical equations available in the literature, extrapolated to the middle of the near-bed layer following the Rouse profile, which in 2DH mode means the middle of the water column. The adopted methodology was described previously in Franz et al. (2017), converging for different numbers of vertical layers. The differences obtained in 2DH or 3D mode should be stressed in simpler test cases to avoid the influence of other mechanisms of sediment transport (e.g., undertow), which would not allow a clear conclusion. Thus, this aspect should be investigated in future studies.

The MOHID modelling system is coupled to the General Ocean Turbulence Model (GOTM) for the vertical turbulent closure (e.g., K-$\varepsilon$ model). Effects of wave breaking on vertical turbulence can be taken into account through surface boundary conditions (Delpey et al., 2014). This mechanism was disregarded in this work, as the idea here was only to provide an approximate representation of the vertical distribution of wave momentum, in order to generate a general undertow pattern. Thus, the corresponding results should be considered as a first qualitative evaluation of the effect of such an undertow in our morphological model, the latter being our focus here. It is left for further work to use a more advanced formulation of 3D wave-current interactions for more quantitative investigations.

The lateral friction can be computed in MOHID considering a null tangential velocity when the cell face is not covered (Leitão, 2003). The slope effects are included in the bedload formulation (see Franz et al., 2017). Alternatively, the bed slope correction was used to represent the seaward sediment transport due to undertow currents, which

cannot be simulated by 2DH models.

[revised manuscript text omitted]

---

## Author Response (AR2)

Dear Editor John M. Huthnance,

The authors are thankful for your comments to the manuscript improvement. The suggestions pointed out in the use of English were implemented. Furthermore, we incorporated our responses to the reviewers into the manuscript as suggested.

Regarding the question about equation 8, due to the shoaling effect, the wave asymmetry increases at the same time that the wavelength decreases. This approach for computing the wave's asymmetry was suggested by Soulsby and Damgaard (2005).

The referenced paper (Franz et al., 2017) is available online and will be published in the next volume of the journal (which looking for the other years may be only in September).

[revised manuscript text omitted]

---

## Author Response (AR3)

Dear Editor John M. Huthnance,

There are two different issues regarding wave asymmetry that may be clarified in equation 8.

First, it has been often reported that wave asymmetry increases when waves propagate shoreward, although wavelength decreases due to shoaling. Indeed in shallow water, asymmetry increases due to nonlinearities generated by wave-bottom interaction. This effect is taken into account in equation 8, as the asymmetry factor depends strongly on the water depth h (not just on the wavelength). The water depth h is decreasing rapidly shoreward and thus the term 1/sinh(kh) is increasing shoreward with decreasing h.

Second, at a given depth, wave asymmetry tends to be higher for longer incident waves, despite the decrease of L due to shoaling. As a physical explanation for this tendency, we think that if incident waves are longer, they propagated over a larger distance in limited water depth (from L/2) than shorter waves. So they may be more affected by nonlinearities, and as a consequence, more asymmetric than shorter waves considered at the same depth. However, as different contributions compete in a complex way when waves propagate in the nearshore, such an explanation is only a hypothesis here.

Anyway, equation 8 appears consistent with the fact that in shallow water the ratio of the second order and the first order velocity $U_{w,2}/U_{w,1}$ should tend to the Ursell number: $Ur \approx HL^2/h^3$ (as mentioned for instance in Dronkers 2005). Examining equation 8 with kh << 1 we have sinh(kh) ≈kh, and thus $\gamma_w \approx$ Ur. Other research works found a relationship between the Ursell number and wave asymmetry (e.g., Ruessink et al., 2012), with the following tendency: for small Ursell numbers asymmetry approaches 0, whereas for larger Ursell numbers asymmetry increases and tends to a maximum asymptotic value. As for a given value of h the Ursell number Ur is increasing with increasing L, this is consistent with an increase of asymmetry with L.

References

Dronkers, J., 2005. Dynamics of coastal systems. Advanced Series on Ocean Engineering 25.

Ruessink, B.G., Ramaekers, G. and Van Rijn, L.C., 2012. On the parameterization of the free-stream non-linear wave orbital motion in nearshore morphodynamic models. Coastal Engineering, 65, pp.56-63.

[revised manuscript text omitted]